# Facile Fabrication of Multi-Structured SiO_2_@PVDF-HFP Nanofibrous Membranes for Enhanced Copper Ions Adsorption

**DOI:** 10.3390/polym10121385

**Published:** 2018-12-13

**Authors:** Haohong Pi, Rui Wang, Baona Ren, Xiuqin Zhang, Jing Wu

**Affiliations:** Beijing Key Laboratory of Clothing Materials R&D and Assessment, Beijing Engineering Research Center of Textile Nanofiber, School of Materials Science and Engineering, Beijing Institute of Fashion Technology, Beijing 100029, China; haohongpi@163.com (H.P.); clywangrui@bift.edu.cn (R.W.); 15231119135@163.com (B.R.)

**Keywords:** electrospun, multi-structured, nanofibrous membrane, superhydrophilic, copper ion (Cu(II)) adsorption

## Abstract

The low-cost, heavy metal ion (Cu(II)) adsorptive multi-structured nanofibrous membranes of silicon oxide naonoparticles in-situ anchored polyvinylidene fluoride-hexafluoropropylene (SiO_2_@PVDF-HFP) fibers were fabricated by the facile electrospinning technique combined with sol–gel strategy. To explore the benefits of the structure-related Cu(II) adsorption capacity, the fiber diameters of SiO_2_@PVDF-HFP nanofibrous membranes were changed which also resulted in the change of their porosity. Taking advantage of the constructed multi-structures and efficient fiber morphology regulation which not only changed the PVDF-HFP nanofibrous membrane from hydrophobic to superhydrophilic but also increased the porosity of the membrane, the SiO_2_@PVDF-HFP nanofibrous membrane with a smaller diameter and a larger porosity exhibits higher Cu(II) adsorption capacity. The adsorption amount was approximate to 21.9 mg per gram of the membrane, which was higher than that of membranes with larger fiber diameter (smaller porosity) and the smooth one. Furthermore, the model isotherms of Freundlich and Langmuir, as well as the kinetic models of pseudo-first-order and pseudo-second-order were preferred to analyze the adsorption equilibrium data. The Freundlich model and the pseudo-first-order were well fitted to the adsorption experimental data. It not only uncovers the structure-related-property of multi-structured nanofibrous membranes, but also provides an efficient and facile way to design heavy metal ion adsorption materials.

## 1. Introduction

Heavy-metal ions are a severe environmental problem in that organic and toxic waste in water not only poses a major health risk to humankind, but also destroys the circulation of the ecological environment [1,2]. Fully cognizant of its hazards, researchers have developed a great deal of techniques to transfer and adsorb heavy metal ions in aqueous solution, such as chemical treatments [3,4,5,6,7,8], biochemical processes [9,10], physicochemical treatments [11,12], et al. Among these, adsorption technology was considered as the simple operation and much more efficient method for removal of heavy metal ions [13,14,15]. Accordingly, various adsorbent such as activated carbon, oxide minerals, and biosorbents were investigated [16,17]. However, what we cannot ignore is that some of these processes have major drawbacks due to high energy consumption and increased operational cost to overcome the additional waste to be disposed of. For example, there is an additional cost to recover the powders that had been used to remove heavy ions in water.

During recent years, nanofibrous membrane has been a promising and highly recommended attractive heavy metal ion adsorbent due to its high surface to volume ratio, and larger specific surface area. Meanwhile, electrospinning is widely recognized as a simple top-down method of fabricating fibrous materials with diameters from tens of nanometers to submicrometers by stretching a viscous conductive fluid into a thin jet via electrostatic force [18,19,20]. Furthermore, the electrospinning method also possesses powerful structural tunability that could create fibers with micro/nanoscale hierarchical structures [21]. Electrospun nanofibrous membrane consists of a three-dimensional interconnected fibrous structure and pores which provide huge specific surface area and porosity, and are easy to recovery after usage, exactly meeting the requirements to prepare materials with efficient heavy metal ion adsorption. In addition, electrospun hierarchically complex structured materials, especially on the micro-/nanometer scale have many prominent applications because they offer larger specific area and additional heterogenous interfaces, which play important roles in enhancing the nano-sized effect. They are competitive in many areas than the bulk or the same sized materials without hierarchical structures [22,23,24].

Herein, the commercial polyvinylidene fluoride-hexafluoropropylene (PVDF-HFP) nanofibrous membranes were fabricated by facile electrospinning. Silicon oxide (SiO_2_) was prepared by Stöber method [25] and in-situ anchored on the PVDF-HFP during its fabrication process, forming multi-structured electrospun superhydrophilic SiO_2_@PVDF-HFP nanofibrous membrane, and used as the adsorbent of heavy metal ion Cu(II) in aqueous solution. Furthermore, to find the benefits of the structure-related properties of Cu(II) adsorption capacity, i.e., the relationship between different morphology of multi-structured nanofibrous membrane and the Cu(II) removal capacity, fiber diameters of SiO_2_@PVDF-HFP were regulated, which also result in the change of the porosity of the fibrous membrane. By controlling the diameter and creating SiO_2_ “bulges” morphology on the PVDF-HFP fibers, greater affinity to aqueous solution, and higher Cu(II) adsorption capacity can be obtained. Such work not only explores the structure-related-property of multi-structured nanofibrous materials, but also provides an efficient way to design the heavy metal ion adsorption materials.

## 2. Materials and Methods

*Preparation of polyvinylidene fluoride-hexafluoropropylene (PVDF-HFP) fibers:* PVDF-HFP (*M*_w_ = 455,000, Sigma-Aldrich, St. Louis, MO, USA) was dissolved in DMF (Beijing Yili Fine Chemical Co., Beijing, China) by stirring for 8 h to form 20 wt %, 22.5 wt %, 25 wt % and 27.5 wt % electrospun precursor solutions. Nearly 4 mL of precursor solution was placed in a 5 mL syringe, and the flow velocity was controlled to 1.0 mL/h by the injection pump. In order to form PVDF-HFP fibers with different diameters, different metal needles of 0.6 mm, 0.8 mm, 1.0 mm, and 1.2 mm inner diameter were equipped. A stainless-steel mesh was covered on the drum (200 r/min) as the collector. The distance between the needle tip and collector was 15 cm, and the voltage was set as 15~20 kV. The obtained PVDF-HFP fibrous membranes were named as PVDF-6G, PVDF-8G, PVDF-10G, and PVDF-12G, respectively.

*Anchoring SiO_2_ nanoparticles on electrospun PVDF-HFP fibrous membrane:* The as-prepared PVDF-HFP fibrous membranes were cut into squares (2.5 cm × 2.5 cm), ultrasonically cleaned by water and ethanol. The washed fibrous membrane was then immersed into 80 mL ethanol. Silicon dioxide nanoparticles (SiO_2_) were prepared according to the Stöber method [25]. Five mL Tetraechoxysilane (TEOS, Sigma-Aldrich, St. Louis, MO, USA) was slowly dropped into the stirred ethanol impregnated with PVDF-HFP fibrous membrane. Ammonium hydroxide was used to change the pH value to 8.5. The above solution was stirred for 18 h. Then the piece of PVDF-HFP membrane was taken out, washed by ethanol to wipe off the residual SiO_2_ nanoparticles, and dried at 60 °C for 2 h. Instruments and Characterization: SEM images were taken by scanning electron microscope (JSM-6700F, Tokyo, Japan). The diameter of the fibers was measured according to SEM images. Contact angles were measured on OCA 20 contact-angle system (Dataphysics, Stuttgart, Germany) at 25 °C. Three μL deionized water droplets were dropped onto the fibrous membrane surface. The average contact angle value was obtained by measuring at five different positions of the same sample. The water adhesion forces were measured using the high-sensitivity micro-electro-mechanical balance system (Data-Physics DCAT11, Stuttgart, Germany). For the PVDF-HFP fibrous membrane, the water droplet (5 μL) was suspended with a metal cap and controlled to contact with the surface of the membrane with a constant speed of 0.005 mm·s^−1^, and then controlled to leave. The force-distance curve was recorded at the real-time. For the SiO_2_@PVDF-HFP, the water droplet was controlled to contact the membrane surface and keep contacting until it was adsorbed. The force-time curve was recorded. The chemical compositions of fibrous membranes were characterized by Fourier Transform Infrared spectrometry (FTIR) on Nicolet 8700 FT-IR spectrometry (Madison, WI, USA) in the wave range from 400 to 4000 cm^−1^. X-ray diffraction (XRD) (Bruker, Karlsruhe, Germany) were collected on a Rigaku-D/max 40,000 V X-ray diffractometer equipped with Cu Kα radiation (λ = 0.15418 nm) at a step width of 5 °/min. The concentrations of Cu(II) solutions were measured by inductively coupled plasma atomic emission spectrophotometer (ICP-AES) (CIROS EOP, Kleve, Germany)

*Cu(II) removal experiments:* A series of adsorption experiments were performed to investigate the efficiency of the multi-structured SiO_2_@PVDF-HFP for the removal of Cu(II) from aqueous solutions. The Cu(II) solutions with different concentrations were prepared by dissolving CuSO_4_·5H_2_O (Beijing Yili Fine Chemical Co., Beijing, China) in deionized water. (1) to determine the influences of pH and adsorption time on removing Cu (II), the pH values of CuSO_4_ solutions were adjusted to 2, 3, 4, 5, and 6, respectively, by 1 mol/L NaOH (Beijing Yili Fine Chemical Co., Beijing, China). 30 mg of SiO_2_@PVDF-HFP fibrous membranes were added into 30 mL of CuSO_4_ aqueous solution (100 mg/L) and then measured concentrations of the Cu solutions after 120 min. (2) in order to explore the equilibrium adsorption kinetics, 30 mg of as-prepared adsorbent was immersed into 100 mL of CuSO_4_ aqueous solution (100 mg/L) at pH = 4.5, followed by sampling the solution after 5, 10, 20, 30, 50, 80,120, 200, 300 min for the analysis of Cu content in the solution. Thus, the adsorption kinetic model was determined. (3) 30 mg of multi-structured SiO_2_@PVDF-HFP fibrous membrane was immersed into 30 mL CuSO_4_ aqueous solution (100 mg/L) with different initial Cu(II) concentrations (30, 50, 70, 100, 200 mg/L) at room temperature. (4) adsorption capacity of SiO_2_@PVDF-HFP with different fiber diameters was compared. When the adsorption reached the equilibrium (adjusted pH = 5 and 6), the Cu(II) content in the solution was measured. The adsorption capacity of the membrane for the Cu(II) removal was calculated as follows:(1)$$q_{e}=\frac{\left(C_{0}-C_{e}\right)V}{m}$$
where *q_e_* is the equilibrium adsorption quantity (mg/g), *C*_0_ and *C*_e_ represent the initial and equilibrium Cu(II) concentration (mg/L), respectively. *V* is the volume of the CuSO_4_ aqueous solution (L), and *m* is the quality of membrane (g).

## 3. Results and Discussion

### 3.1. Fabrication and Morphology of Multi-Structured SiO_2_@PVDF-HFP Nanofibrous Membranes

Polyvinylidene fluoride-hexafluoropropylene (PVDF-HFP) was used as the raw materials to fabricate the nanoparticle anchored on nanofiber multi-structured membranes. Figure 1 schematically illustrates the membrane fabrication process. An electrospinning technique was used to process (PVDF-HFP)/*N*,*N*′-dimethylformamide (DMF) solution into a fibrous membrane. The obtained PVDF-HFP fibrous membrane was then immersed in the stirring alcohol, and TEOS was added dropwise at the same time. Since the electrospun PVDF-HFP nanofibrous membranes act as the skeleton, during the process of synthesis SiO_2_ by Stöber sol-gel method [25], SiO_2_ nanoparticles had been in-situ anchored on their surfaces.

In this work, the pore size of both the PVDF-HFP and SiO_2_-PVDF-HFP membranes were controlled by regulating the fiber diameters during the electrospinning process, mainly by adjusting the concentrations of electrospinng solution as well as the inner diameters of electrospun nozzles. As we know, the most straightforward means of increasing the pore size of electrospun scaffolds involves careful modification of system or process parameters during the fabrication process. For one thing, smaller pores can be achieved by decreasing the fiber diameters, a technique that is described by a number of groups as means of decreasing the pore size of scaffolds electrospun from both synthetic and natural polymers [26,27,28,29]. For the other thing, statistical modeling predicts a relationship between the fiber diameter and the pore size of electrospun scaffolds, where a smaller fiber diameter correlates with an increase in the pore size [24,30]. What’s more, our previous works [31,32] also proved that the pore size of micro/nano fibrous membrane can be controlled by regulating the electrospinning parameters. Images in Figure 2 are the scanning electron microscopy (SEM) of as-prepared samples. To obtain different fibers’ diameter, different metal needles with the inner diameters of 0.6 mm (6G), 0.8 mm (8G), 1.0 mm (10G), and 1.2 mm (12G) were equipped during the fabrication process. Accordingly, Figure 2a–d are the typical SEM images of electrospun PVDF-HPF fibers with different fiber diameters, which are named as PVDF-HFP-6G, PVDF-HFP-8G, PVDF-HFP-10G, and PVDF-HFP-12G, respectively. The statistic average diameters are 150 nm, 201 nm, 214 nm, and 236 nm, respectively.

After 18 h of the sol-gel fabrication process, SEM images of SiO_2_ nanoparticle anchored samples are exhibiting in Figure 2e–h (named SiO_2_-PVDF-HFP-6G, SiO_2_-PVDF-HFP-8G, SiO_2_-PVDF-HFP-10G, and SiO_2_-PVDF-HFP-12G). It can be seen that the SiO_2_ nanoparticles are successfully fabricated and anchored on the PVDF-HFP fibrous membranes (orange dotted frames in Figure 2e–h). Meanwhile, for both the PVDF-HFP and SiO_2_@PVDF-HFP samples, the fibers are randomly oriented on the scaffold, and fibers’ diameters are increased by increasing the nozzles’ inner diameters. Compared with PVDF-HFP fibers, the fiber diameters of SiO_2_@PVDF-HFP increased, which can result from the coating of sol during the SiO_2_ fabrication process via the Stöber method.

### 3.2. Wettability Characterization

Figure 3a exhibits the relationship between electrospun nozzles type, solution concentrations and fiber diameter. With the increase of the inner diameter of electrospun needles as well as the electrospun solution concentration, the fiber diameter of the obtained PVDF-HFP fibers increased. The wettability of both PVDF-HFP and SiO_2_@PVDF-HFP fibrous membranes were carried out by measuring the water contact angles (WCA). A water droplet about 3 μL allowed to contact the surface of all as-prepared samples (Figure 3b). Water contact angles are 151.2° ± 0.6°, 147.7° ± 0.5°, 144.0° ± 0.7°, and 141.5° ± 0.9° corresponding to PVDF-HFP-6G, PVDF-HFP-8G, PVDF-HFP-10G, and PVDF-HFP-12G, respectively, indicating the PVDF-HFP nanofibrous membranes are hydrophobicity (blue line and inserted photos with dotted frames in Figure 3b). The PVDF-HFP natural property and porous membranes with high surface roughness are the main reasons for its hydrophobicity. Meanwhile, it also can be seen that the different diameter shows tiny influence to the wettability. With the increasing of fiber diameter, the WCA of PVDF-HFP fibrous membranes decreased slightly, attributing to that for a fibers-composed surface, the smaller fiber diameter usually brings about greater surface roughness, which can enhance or “enlarge” the wetting behavior [33,34]. In contrast, due to abundance of hydroxyl groups of hydrophilic SiO_2_ nanoparticles prepared via Stöber method, when 3 μL water droplet dropped on SiO_2_@PVDF-HFP membrane surfaces, it immediately spreads on the membrane with the contact angle near to 0°, exhibiting superhydrophilic property (rose line and inserted photos with dotted frames in Figure 3b).

The heavy metal ion removal process was in an aqueous solution system. Such a reaction process can be deemed as the interfacial reaction in which the interaction between the aqueous solution and the membrane plays important role. Accordingly, the drawing force of the porous membranes towards the water droplet was measured by using a micro-force balance system. Two testing models were employed: in Model 1, a water droplet (5 μL) was brought to contact the membrane and then pulled off, which was used to probe the force between the water droplet and the hydrophobic PVDF-HFP porous membrane (schematic illustration on top of Figure 3c). While, in Model 2, the water droplet was brought to contact the membrane without pulling off, from which water spreading or adsorbing time by the superhydrophilic SiO_2_@PVDF-HFP porous membrane can be measured, and the real-time force change was recorded (schematic illustration on top of Figure 3d).

As shown in Figure 3c, when water droplets contacted the hydrophobic PVDF-HFP porous surface and then moved downward, before being fully separated from fibrous surface, the force gradually increased with increasing the “moving downward” distance (D). At the time when the water droplet separated from the membrane, the force reached a maximum value (F) after which it then decreased abruptly. The recorded corresponding force for the porous membrane PVDF-HFP-6G with smallest fiber diameter was 54.88 μN, which was higher than that of PVDF-HFP-8G (49.98 μN), PVDF-HFP-10G (32.34 μN), and PVDF-HFP-12G (31.36 μN). It indicates that the nanofibrous membrane PVDF-HFP-6G exhibits higher adhesive force to water which is in favor to adsorb solution in further sol-gel SiO_2_ fabrication process. Figure 3d shows the testing results of superhydrophilic SiO_2_@PVDF-HFP porous membranes. It should be pointed out that for the superhydrophilic porous surface, once the water droplet contacted on its surface, it can be adsorbed immediately. The maximum force value appears when the water droplet contacted on the membrane and drops rapidly in a short time, regarded as a shorter time to reach the peak suggesting faster water spreading or adsorbing. The recorded time was 13.60 s, 14.88 s, 18.60 s, and 18.35 s for SiO_2_@PVDF-HFP-6G, SiO_2_@PVDF-HFP-8G, SiO_2_@PVDF-HFP-10G, and SiO_2_@PVDF-HFP-12G, respectively. That is, the multi-structured SiO_2_@PVDF-HFP-6G nanofibrous membrane shows stronger and faster absorbed affinity to aqueous solution, which contributes to the interaction between the ions in aqueous solution and the porous membrane surface.

### 3.3. FTIR and XRD Analysis

The Fourier Transform Infrared Spectroscopy (FTIR) was used to characterize the chemical compositions of as-prepared PVDF-HFP, SiO_2_@PVDF-HFP electrospun nanofibrous membrane, and the synthetic inorganic silica nanoparticles (Figure 4a). It was carried out in the range of 400 cm^−1^–4000 cm^−1^ to analysis and confirm the presence of functional groups [35]. Compared with the spectra of PVDF-HFP porous membrane, peaks in the range of 3639–2730 cm^−1^ appeared, which can be attributed to the anti-symmetric stretching vibration peaks of the silanol (Si–OH) group for the SiO_2_@PVDF-HFP membrane [36,37,38]. The peak around 1630 cm^−1^ is ascribed to the bending vibration of O–H groups [37]. The presence of SiO_2_ is verified by absorption bands at 1135 to 990 cm^−1^ [38,39], Si–O–Si anti-symmetric stretching vibrations, and a very weak Si–OH bending vibration absorption peak near 944 cm^−1^ [39,40]. Since Si–OH is susceptible to thermal dehydration to form Si–O–Si, almost no visible Si–OH absorption peak can be detected in the spectrum. In addition, peaks in the range of 852–769 cm^−1^ and 510–418 cm^−1^ are represented the Si–O–Si asymmetric stretching vibration and bending vibration, respectively [39,41]. Such characteristic peaks of SiO_2_ nanoparticles appeared in the SiO_2_@PVDF-HFP spectrum proving that silica nanoparticles successfully loaded on the fibers’ surface.

To analysis the crystal property of as-prepared samples, Figure 4b exhibits the X-ray diffraction (XRD) spectrum. For SiO_2_ nanoparticles, XRD peaks appeared in the low diffraction angle region near 2θ = 23°, indicating that the obtained SiO_2_ nanoparticles are amorphous [38,42]. Once 2θ over 23°, the diffraction intensity gradually decreased with slightly fluctuations, and finally tended to become smooth. Making a comparison with PVDF-HFP and SiO_2_ nanoparticles, the diffraction spectrum of the SiO_2_@PVDF-HFP sample shown a typical SiO_2_ amorphous diffraction peak near 2θ = 20°, which can also be seen in the two-dimensional diffraction patterns (insertions in Figure 3b), indicating that the amorphous SiO_2_ nanoparticles anchored on PVDF-HFP nanofibrous membrane [39,43,44].

### 3.4. Copper Ion (Cu(II)) Adsorption in Aqueous Solution

In addition, batch adsorption experiments were carried out to study the adsorption and removal capacity of multi-structured SiO_2_@PVDF-HFP electrospun nanofibrous membranes. Copper ions (Cu(II)) in aqueous solution acts as a “target” degradation for adsorption. On account of the substantial verified electrostatic/adsorption properties of an adsorbent with the solution pH value, it is known that the initial pH value exhibits a significant effect on adsorption capacity of metal ions [45,46]. To explore the optimized pH value, the effect of solution pH values (from pH = 1.0 to 6.0, since the precipitations of copper hydroxide would form at the solution pH values above 7) [47] on the adsorptions of Cu(II) ion were investigated (Figure 5a). When the pH value is in the range of 1 to 5, the Cu(II) simply adsorbed. Once the pH changed to 6, the adsorption amounts were increased abruptly. It can be attributed to that the surface of the adsorbent had positive charges at low pH values (1–5). When the pH values were raised over 5, the –OH groups on the SiO_2_ nanoparticle surfaces suggested making the adsorbent membrane surface more negatively charged [48]. Meanwhile, the incorporation of negative functional groups including siloxane, silanol, and amine groups into the SiO_2_@PVDF-HFP fibrous membrane which enhanced the available active sites for adsorption process [48,49]. For one thing, the Cu(II) metal ions with positive charges can easily interact with the surface of electrospun membrane. Cu(II) would hydrolyze to Cu(OH)_2_, Cu(OH)^+^, and Cu_2_(OH)_2_^2+^ when the pH over 5 which will promote the adsorption [47,50]. Whereas, further increase in the pH level once at high pH, i.e., pH > 7 (alkaline solutions), the metal cations start to react with hydroxide ions to form metal hydroxide and get precipitated.

In Figure 5b, the effect of initial concentration of Cu(II) ions on the adsorption capacity was also depicted. The relationship between initial Cu(II) concentration and the removal rate by multi-structured SiO_2_@PVDF-HFP nanofibrous membrane can be obtained according to the slope in the plot of adsorption amount (Q (mg/g)) versus initial Cu(II) concentration (C (mg/L)). The overall trend exhibits that the adsorption capacity of SiO_2_@PVDF-HFP electrospun membrane increased with the increasing of initial Cu(II) concentrations. However, it can be seen that the slope of the Q~C plot was larger when the initial Cu(II) concentration was below 50 mg/L than once it was over 50 mg/L, exhibiting that the multi-structured nanofibers exhibit a higher removal rate at relatively lower initial concentrations. At first, plenty of Cu(II) were adsorbed on the surface of the membrane at any initial concentrations by the coaction of both the active sites of fibers and the roughness of the multi-structured membrane. With the increase in initial Cu(II) ion concentration, the ion coordination sites on the surface of the nanofiber membrane were rapidly occupied [45]. Taking advantages of the roughness of the obtained multi-structured fibrous surface, plenty of Cu(II) ions can only be adsorbed on the multi-structured nanofiber surface [51]. The adsorption of Cu(II) ions can only occupy on the rough areas of the membrane, which cause the Cu(II) adsorption rate to decrease.

Furthermore, the adsorption time is also one of the most important factors to evaluate the removal performance of heavy metal ions [45], and its influence on adsorption capacity of multi-structured SiO_2_@PVDF-HFP-6G fibrous membrane in Cu(II) aqueous solution are shown in Figure 5c. It exhibited a relatively high adsorption rate during the early stage of the adsorption process, then reduced sharply with the increasing of adsorption time, and finally reached the equilibrium time of 120 min. During the whole process, the concentration of Cu(II) in aqueous solution decreased from 110 mg/L to 99.02 mg/L, and keep constant after 120 min.

The unique characteristics of electrospun fibrous material are the very high surface area to volume ratios, high porosity that results in high surface cohesion, making it an excellent candidate for adsorption. By adjusting the fiber diameter, the morphology of fibrous membrane can be easily regulated, which influences the porosity of the fibrous membrane. As showing in Figure 5d, the Cu(II) adsorption capacity of multi-structured SiO_2_@PVDF-HFP electrospun nanofibrous membranes with different fiber diameters as well as the SiO_2_ anchored smooth PVDF-HFP (prepared via sip-coating, named SM) were measured under the conditions of a pH value of 6, and an initial Cu(II) ion concentration of 50 mg·L^−1^. The adsorption efficiency gradually decreased corresponding to the membranes 6G, 8G, 10G, and 12G. The SiO_2_@PVDF-HFP-6G exhibits the highest Cu(II) adsorption capacity (21.9 mg per gram of membrane). Meanwhile, the adsorption capacity of the SM membrane was much lower than that of the other multi-structured electrospun SiO_2_@PVDF-HFP membranes which, mainly due to that the rough fibrous membranes, have larger surface areas providing more available active sites which facilitated the adsorption of Cu(II) metal ions [39,45,51]. In addition, the relation between the average fiber diameter and the porosity were measured via anhydrous ethanol uptake, and the results were shown in Figure 5e. For both the SiO_2_@PVDF-HFP and PVDF-HFP membranes, the porosity decreased with the decreasing of the average fiber diameter. The porosity of SiO_2_@PVDF-HFP-6G, SiO_2_@PVDF-HFP-8G, SiO_2_@PVDF-HFP-10G, and SiO_2_@PVDF-HFP-12G were 89.74%, 83.32%, 80.54%, and 74.69%, respectively, which are higher than that those of PVDF-HFP membranes. Thus, it can be seen that for one thing, the constructed multi-structures offer larger porosity. For the other thing, the average fiber diameter of multi-structured nanofibrous membranes plays an important role in determining the porosity as well as affects the adsorption of Cu(II), i.e., for the membranes with similar morphology, the smaller fiber diameter caused larger porosity, which enhanced the Cu(II) adsorption in aqueous solution.

The recyclable property of multi-structured SiO_2_@PVDF-HFP-6G nanofibrous membrane was studied by repeating the Cu(II) removal experimental process five times by using the same nanofibrous membrane. The regeneration of Cu^2+^ metal ions by the membrane was treated with 50 mL hydrochloric acid solution at pH = 1. After 2 h stirring at 25 °C, these membranes were rinsed with diluted alcohol and deionized water repeatedly. As shown in Figure 5f, the adsorption amount of obtained nanofibrous membrane after five times recycle usage was 18.4 mg/g, which exhibits only slight decrease from that of adsorption amount when used for the first time (21.9 mg/g), showing a superior potential for recycle application.

### 3.5. Adsorption Kinetics and Isotherms Simulation

To better understand the adsorption ability of the multi-structured SiO_2_@PVDF-HFP nanofibrous membranes toward Cu(II) metal ions, the adsorption experimental data are normally used to evaluated by the kinetic models. The kinetic models usually consist of pseudo first-order adsorption model and pseudo second-order adsorption model [52,53,54,55], both of which can be expressed by kinetic equations:

Pseudo first-order adsorption model:(2)$$\ln\left(q_{e}-q_{t}\right)=\ln q_{e}-k_{1}t$$
where *q_t_* is the amount of Cu(II) adsorbed (mg/g) at time *t*, *q_e_* is the amount of Cu(II) adsorbed (mg/g) at equilibrium time, respectively, *k*_1_ is the rate constant (1/min) of pseudo first-order adsorption.

Pseudo second-order adsorption model:(3)$$\frac{t}{q_{t}}=\frac{1}{k_{2}q_{e}^{2}}+\frac{t}{q_{e}}$$
where *k*_2_ (g/mg/min) represents the rate constant of pseudo-second-order reaction. The value of *q_e_* and *k*_2_ can be determined by the slope and intercept of the straight line of the plot of “*t*/*q_t_* versus *t*”.

The first-order kinetic equation means that the adsorption rate is linear with a reactant concentration, while the second-order kinetics means adsorption rate is linear with two reactants. In this work, the applicability of suitable kinetic model can be evaluated through the magnitude of the so-called correlation coefficients, i.e., *R*^2^. Generally, the higher *R*^2^ represents better applicability of the corresponding kinetic model [52,54]. Both the pseudo-first-order adsorption model and the pseudo-second-order adsorption model were applied to fitted the adsorption behavior. As shown in Figure 6a,b, the pseudo-second-order kinetic model (*R*^2^ = 0.99735) was better described the kinetic data of Cu(II) removal process in this work. More details can be seen in Table 1. The *q_e_* values (*q_e(cal)_* = 10.05025 mg/g) calculated from the pseudo-second-order model for Cu(II) metal ions adsorption are closer to the experimental data (*q_e(exp)_* = 9.9818 mg/g), indicated the adsorption process of the multi-structured electrospun SiO_2_@PVDF-HFP fibrous membrane on Cu(II) ions is more consistent with the pseudo-second-order kinetics [56].

Isothermal adsorption models are also used to evaluated the adsorption simulation [54], which consist of Langmuir adsorption model and Freundlich adsorption model. Based on the relationship between the initial Cu(II) concentrations with the adsorption capacity of multi-structured electrospun SiO_2_@PVDF-HFP-6G membrane shown in Figure 5b, the Freundlich and Langmuir isotherm models were used to fit the equilibrium. The isotherm equations are expressed as follows [54,57,58].

Langmuir isotherm model:(4)$$\frac{C_{e}}{q_{e}}=\frac{1}{q_{m}K_{L}}+\frac{C_{e}}{q_{m}}$$
where *q_e_* is the amount of the metal ions adsorbed (mg/g), *q_m_* is the maximum adsorption capacity (mg/g), *C_e_* is the equilibrium concentration of the metal ions (mg/L), and *K_L_* is the Langmuir constant related to the free energy of adsorption (L/mg). The values of *q_m_* and *K_L_* can be derived from the slope and intercept of *C_e_/q_e_* versus *C_e_*.

Freundlich isotherm model:(5)$$\ln q_{e}=\frac{1}{n}\ln C_{e}+\ln K_{F}$$
where *q_e_* is the equilibrium adsorption capacity (mg/g), and *K_F_* and *n* are Freundlich constants related to the adsorption capacity and adsorption intensity, respectively. If 1/*n* is smaller than unity, the adsorption process is favorable. The values of *K_F_* and *n* can be obtained from the slope and intercept evaluated from the plot of ln*q_e_* versus ln*C_e_*.

The Freundlich isotherm model is suitable for the adsorption on heterogeneous surfaces which can describe adsorption data over a restricted range. The Langmuir adsorption model is mainly used for the adsorption process with strong interaction with the adsorption membrane surface, and the adsorption process is single layer adsorption. The isotherm simulation results are shown in Figure 6c,d, and much more details are presented in Table 2. Making comparison of the correlation coefficients, it was found that the equilibrium data was best described by Freundlich isotherm model (*R*^2^ = 0.97377) than that from Langmuir isotherm model (*R*^2^ = 0.92495) [49]. It was caused by the multilayer reaction adsorption of Cu(II) ions onto the heterogeneous surface of multi-structured SiO_2_@PVDF-HFP fibrous membranes, i.e., the stronger binding sites are occupied firstly by the membrane, which is consistent with the mechanism of Freundlich isotherm adsorption [53]. And the Freundlich constants (1/*n*) was smaller than 1 [56], indicating the adsorption of Cu(II) metal ions onto the multi-structured SiO_2_@PVDF-HFP fibrous membranes are the favorable processes.

## 4. Conclusions

In summary, SiO_2_ nanoparticles were successfully anchored on electrospun PVDF-HFP nanofibrous membranes, forming superhydrophilic multi-structured SiO_2_@PVDF-HFP nanofibrous membranes. The benefit of the structure-related Cu(II) adsorption capacity is that the multi-structured SiO_2_@PVDF-HFP nanofibrous membrane with smaller fiber diameter, larger porosity exhibits greater affinity to aqueous solutions and higher Cu(II) adsorption capacity. Taking advantages of constructed SiO_2_ nanoparticles and fiber morphology regulation, not only was the Cu(II) adsorption capacity enhanced, but also the adsorption simulation was fitted well with the kinetic model of pseudo-first-order and the isotherm model of Freundlich. This work uncovers the structure-related-property of multi-structured nanofibrous materials, and provides an efficient and facile way to design the heavy metal ion adsorption materials.

## Figures and Tables

**Figure 1 polymers-10-01385-f001:**
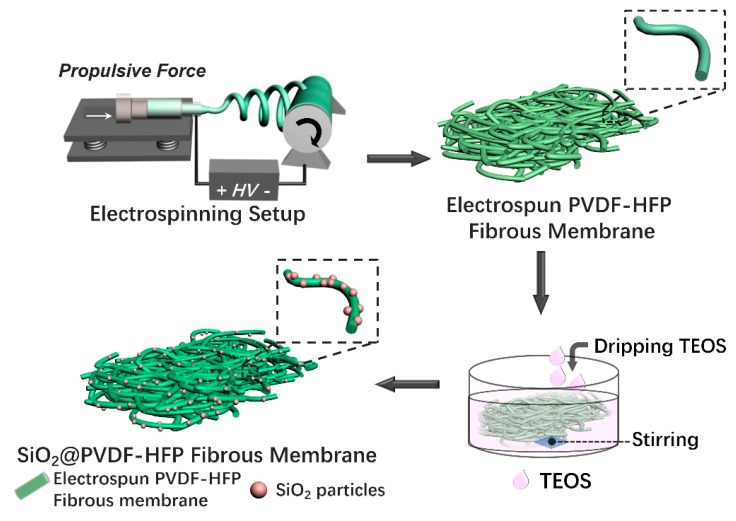
Schematic illustration of multi-structured SiO_2_@PVDF-HFP nanofibrous membrane preparation process. The PVDF-HPF nanofibrous membrane was obtained by electrospinning, and then was immersed in stirred alcohol, TEOS was successively dripped into the alcohol solution. After the minimum time of 6 h, the multi-structured SiO_2_@PVDF-HFP nanofibrous membrane was successfully fabricated.

**Figure 2 polymers-10-01385-f002:**
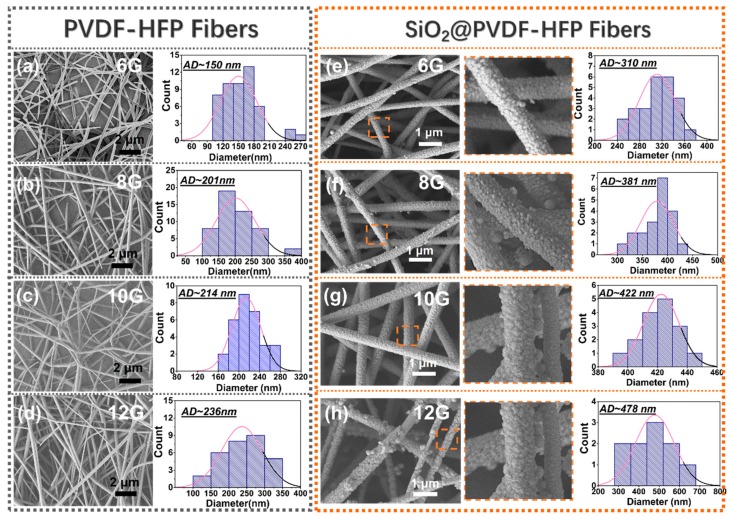
SEM images and statistical analysis of fiber diameters of PVDF-HFP and SiO_2_@PVDF-HFP nanofibrous membranes. (**a**–**d**) PVDF-HFP-6, -8, -10 and -12 electrospun fibrous membranes. The average fiber diameters (AD) are 150 nm, 201 nm, 214 nm, and 236 nm, respectively. (Bar: 2 μm) (**e**–**h**) SiO_2_@PVDF-HFP-6, -8, -10 and -12 electrospun membranes. (Bar: 1 μm). The corresponding enlarged zoomed views were exhibited in the orange dotted frames. SiO_2_ particles were anchored on the fiber surfaces. The average fiber diameters are 310 nm, 381 nm, 422 nm, and 478 nm, respectively.

**Figure 3 polymers-10-01385-f003:**
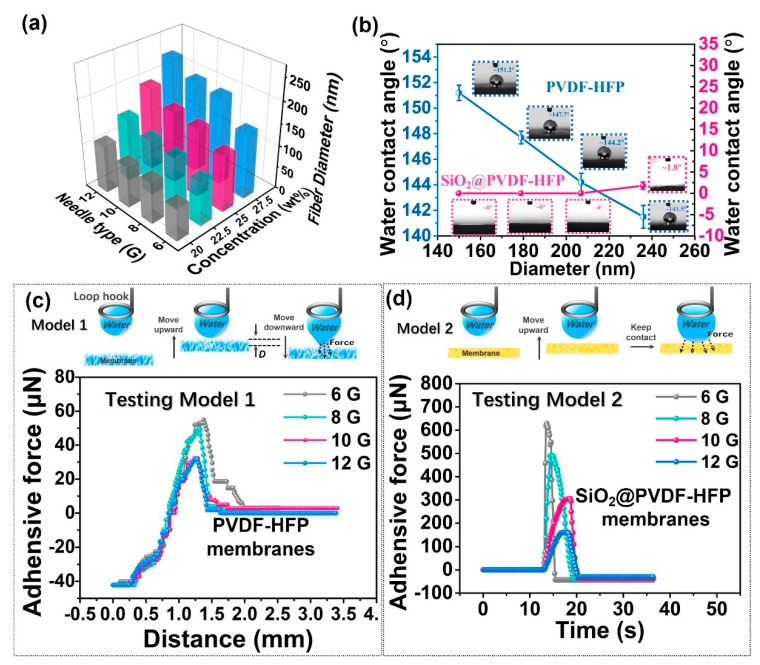
(**a**) The relationship between needle type, concentrations of electrospun solution, and fiber diameter. (**b**) Water contact angles of PVDF-HFP-6G, -8G, -10G, -12G and SiO_2_-PVDF-HFP-6G, -8G, -10G and -12G. (**c**) Water droplet (5 μL) was brought to contact the membrane and then pulled off. It enabled to probe the interaction between water droplet and membrane. Top: schematic illustration of testing model 1. (**d**) Water droplet was brought to contact the membrane without pulling off. The real-time force change was recorded. Top: schematic illustration of testing model 2.

**Figure 4 polymers-10-01385-f004:**
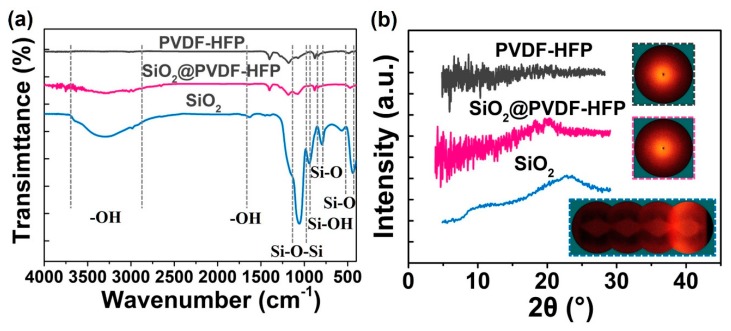
(**a**) FTIR spectrum and (**b**) XRD curves of PVDF-HFP, SiO_2_ nanoparticles, and multi-structured SiO_2_@PVDF-HFP nanofibrous membrane.

**Figure 5 polymers-10-01385-f005:**
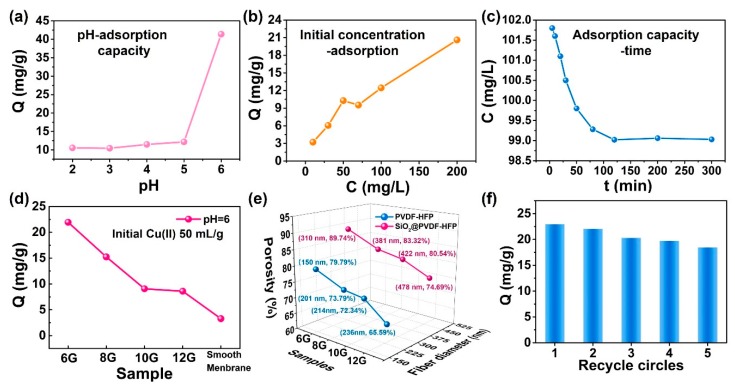
(**a**) The relationship between pH value and Cu(II) removal capacity. (**b**) The change of Cu(II) amount with different initial Cu(II) concentrations. (**c**) Change of the total Cu(II) in the solution with time. (**d**) The removal capacity of as-prepared SiO_2_@PVDF-HFP samples with different fiber diameters. (**e**) Relationships between fiber diameters and porosity of both PVDF-HFP and SiO_2_@PVDF-HFP electrospun nanofibrous membrane. (**f**) The recyclable properties of SiO_2_@PVDF-HFP-6G nanofibrous membrane for Cu(II) adsorption.

**Figure 6 polymers-10-01385-f006:**
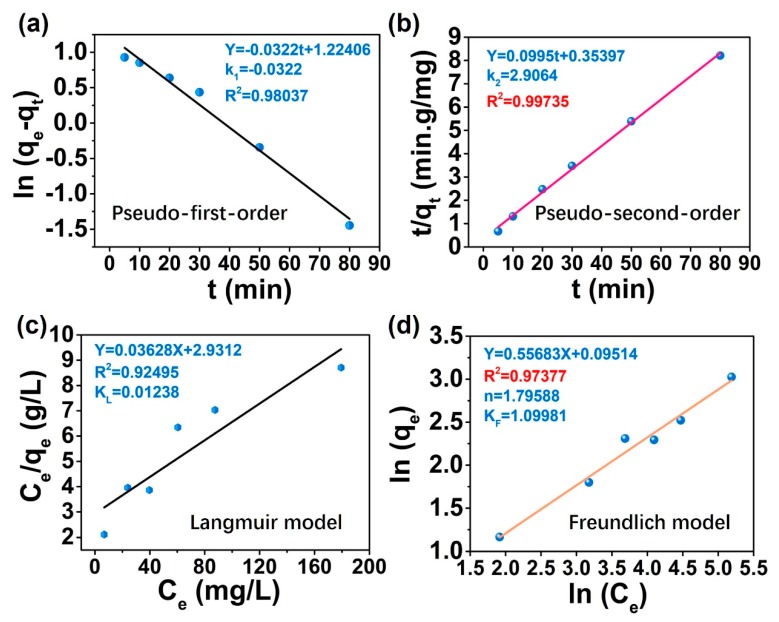
The kinetic fitting model of (**a**) Pseudo-first-order plot and (**b**) Pseudo-second-plot. (**c**) and (**d**) are the Langmuir and Freundlich isothermal models for adsorption of Cu(II) metal ions by the multi-structured SiO_2_@PVDF-HFP membrane.

**Table 1 polymers-10-01385-t001:** Kinetic parameters for the pseudo-first-order and pseudo-second-order models of Cu(II) metal ions adsorption on SiO_2_@PVDF-HFP-6G nanofibrous membrane.

	Pseudo First Order Model			Pseudo Second Order Model		
*q_e(exp)_* (mg/g)	*k* _1_	*R* ^2^	*q_e(cal)_* (mg/g)	*k* _2_	*R* ^2^	*q_e(cal)_* (mg/g)
9.9818	−0.0322	0.98037	3.401	2.9064	0.99735	10.05025

**Table 2 polymers-10-01385-t002:** Parameters for the Langmuir and Freundlich models of adsorption of Cu(II) ions on multi-structured SiO_2_@PVDF-HFP nanofibrous membranes.

Langmuir Isothermal Models			Freundlich Isothermal Models		
*K_L_*	*R* ^2^	*q_m_*	*K_F_*	*R* ^2^	1/*n*
0.01238	0.92495	27.5634	1.09981	0.97377	0.55683

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
