# Peer review of "Facile Fabrication of Multi-Structured SiO2@PVDF-HFP Nanofibrous Membranes for Enhanced Copper Ions Adsorption"

_polymers, 2018, doi:10.3390/polym10121385_

Round 1
Reviewer 1 Report
This paper presents the facile fabrication of multi-structured SiO2@PVDF-HFP nanofibrous membranes for enhanced copper ions adsorption. Generally, the paper is well organized and written. It could be considered for publication if major revisions are made to address the following raised questions:
1. In the Introduction section, the authors talked about the background and work that has been done previously. However, the reviewer did not see the novelty of the work compared to previous work. From the citation [25] in the paper, this method was developed by other researchers. What are the contributions of the work?
2. There are many long paragraphs in the paper. It may be better to put these paragraphs into shorter ones. This will help the readers to easily understand.
3. The authors did some line fitting according to some models and experimental data received. What do the comparisons suggest? What can we learn?
Author Response
Dear Editor,
Please find submitted the revised manuscript entitled “Facile fabrication of multi-structured SiO2@PVDF-HFP nanofibrous membranes for enhanced copper ions adsorption”. We appreciate the constructive and positive comments provided by the reviewers and acknowledge the improvements that these comments have made to the manuscript. The following is a summary of the responses to the comments on this manuscript.
The authors would like to thank the referees for the important suggestions. Our changes/responses to the referee’s comments are listed point-by-point below.
Referees #1
Comment (1): In the Introduction section, the authors talked about the background and work that has been done previously. However, the reviewer did not see the novelty of the work compared to previous work. From the citation [25] in the paper, this method was developed by other researchers. What are the contributions of the work?
Response (1): The citation [25] in our manuscript was propose a method for preparing silica nanoparticles as well as studied the influence of experimental conditions on the particle size control and dispersion. In this work, we mainly focused on the structure-related-property of SiO2 anchored PVDF-HFP nanofibrous membranes, i.e., the fiber diameters and porosity of SiO2@PVDF-HFP were adjusted, and the structured-related Cu(II) removal property was systematic studied, rather than paid much attention to the synthesis of SiO2 nanoparticles. Thanks for the referee’s suggestion, we also made some modifications in the introduction part to highlight the novelty of our work (line 62-64, 67, in revised manuscript).
Comment (2): There are many long paragraphs in the paper. It may be better to put these paragraphs into shorter ones. This will help the readers to easily understand.
Response (2): Thanks for the referee’s kind suggestion. We have put some long paragraphs into shorter ones to make them easily understand. The changed details are indicated in the revised manuscript (line 160, 207, 298, revised manuscript).
Comment (3): The authors did some line fitting according to some models and experimental data received. What do the comparisons suggest? What can we learn?
Response (3): The kinetic models and isothermal adsorption models are normally used to evaluated the adsorption simulation. The kinetic models usually consist of pseudo first-order adsorption model and pseudo second-order adsorption model, both of which can be expressed by kinetic equations. The first-order kinetic equation means that the adsorption rate is linear with a reactant concentration, while the second-order kinetics means adsorption rate is linear with two reactants. In this work, the correlation coefficient (R2) of second-order kinetic is larger, and the qe value calculated by the second-order kinetics is closer to the experimental data, indicated the experiments are more consistent with the second-order kinetic model.
Similarly, there are still two models for the isotherm adsorption research, Freundlich adsorption model and Langmuir adsorption model. The Freundlich isotherm model is suitable for the adsorption on heterogeneous surfaces which can describe adsorption data over a restricted range. The Langmuir adsorption model is mainly used for the adsorption process with strong interaction with the adsorption membrane surface, and the adsorption process is single layer adsorption. In this work, the multi-structured nanofibrous membrane adsorbed Cu2+ ions mostly by physical action in the heterogeneous surface. When the isotherm simulation was applied by both Freundlich and Langmuir models, it fitted the Freundlich model well result from the higher R2 value. Furthermore, we added the relative illustrations in main manuscript.
All changes have been highlighted in the revised manuscript. Thanks for your consideration of the manuscript.
Sincerely,
Jing Wu
E-mail: [email protected]

Reviewer 2 Report
Dear authors,
The presented manuscript covers an interesting field and describes a facile and versatile route to develop membranes for copper adsorption.
The work is well structured and described, the experimental work is sufficiently detailed and the characterization techniques well used and discussed.
However, there are some points that need further clarification to make the paper more sound. The adsorption mechanism of the membranes is measured and the kinetic mechanisms proposed justifying all the experimental data observed. Although these observations validate the proposed route it is not clear how these membranes can be applied in a real environment. Does the presence of other metals affect the efficiency of the membranes? what happens in case of high concentration of Cu? what may happen if other Cu cathions are present?
Would it be possible to these membranes to recover after metal adsorption? this should have been evaluated in the paper.
I would suggest to the authors to discuss all these points in the manuscript to make it more interesting to all potential readers. After these minor modifications the paper should be ready for acceptance in Polymers.
With kidn regards
Author Response
Dear Editor,
Please find submitted the revised manuscript entitled “Facile fabrication of multi-structured SiO2@PVDF-HFP nanofibrous membranes for enhanced copper ions adsorption”. We appreciate the constructive and positive comments provided by the reviewers and acknowledge the improvements that these comments have made to the manuscript. The following is a summary of the responses to the comments on this manuscript.
The authors would like to thank the referees for the important suggestions. Our changes/responses to the referee’s comments are listed point-by-point below.
Referee #2:
Comment (1): Although these observations validate the proposed route it is not clear how these membranes can be applied in a real environment.
Response (1): One of the strategies to achieve applications for nanofibrous materials is solution processable production due to the merit of easy process, large areas, and low cost. Electrospinning technology can meet above kinds of characteristics, exhibiting great advantages and potentials for the real applications. Meanwhile, when the electrospun nanofibrous membrane was used as adsorbent in aqueous solution, compared with powder adsorbent, the great advantage is that it can be effectively recycled after usage, thereby avoiding the possible secondary pollution.
Comment (2): Does the presence of other metals affect the efficiency of the membranes? What happens in case of high concentration of Cu?
Response (2): Indeed, in this work, in order to easily explore the structure-related-property, i.e., the relationship between the different morphologies of nanofibrous membrane and the heavy metal ion removal capacity, we only chosen Cu2+ as the removal target ions. Thanks very much for the referee’s suggestion, we will choose other materials in further related works to make the research more systematically.
Taking into consideration of the concentration of Cu, the effect of initial Cu(II) concentration on the adsorption capacity was shown in Figure 5(b), main manuscript. Accordingly, the optimal initial concentration of Cu(II) can be obtained. Although the adsorption amount of Cu(II) increased with the increasing of Cu(II) concentration, yet when the concentration reaches to a certain threshold, the adsorption amount no longer increased, indicating reaches to the adsorption equilibrium [Sabourian, Vahid. Irani, Mohammad. et, al. RSC Adv., 2016, 6, 40354. Feng, Quan. et, al. J. Hazard. Mater., 2018, 344, 819. Chen, Hong. Et, al. J. Hazard. Mater., 2018, 345, 1. Wang, Panpan. Li, Lili. Et, al. New J. Chem., 2018, 42, 17740.] Thus, in this work, the initial Cu(II) concentration was 50 mg/L, no higher concentrations were chosen.
Comment (3): What may happen if other Cu cations are present?
Response (3): In addition to Cu2+ ions, the presence of copper ions in water also can be as Cu+ and [Cu(NH3)4]2+. However, the electrostatic interaction between Cu+ and water molecular is much smaller than that of Cu2+. Once Cu+ ions exist in water, they are easily changed into Cu2+ and Cu by disproportion decomposition. Meanwhile, [Cu(NH3)4]2+ ions exist in a special ammonia environment, which is actually difficult to achieve in real water environment. Therefore, Cu2+ was chosen as the target ions in this work.
Comment (4): Would it be possible to these membranes to recover after metal adsorption? This should have been evaluated in the paper.
Response (4): Thanks very much for the referee’s kind suggestion. The recyclable property of SiO2@PVDF-HFP nanofibrous membrane after Cu(II) adsorption was evaluated and added to the revised manuscript as Figure 5(f). It can be seen that the adsorption amount of obtained nanofibrous membrane after 5 times recycle usage exhibits only slight decrease than that of initial adsorption amount, showing a superior potential for recycle application.
All changes have been highlighted in the revised manuscript. Thanks for your consideration of the manuscript.
Sincerely,
Jing Wu
E-mail: [email protected]

Round 2
Reviewer 1 Report
The authors addressed the raised questions and the paper could be considered for publication.